# Religious and cultural perspectives on assisted reproductive technology in Ghana: A comparative analysis of traditionalist, islamic, and christian beliefs

Kwadwo Asante-Afari ᴵᴰ*

Health Promotion Division, Ghana Health Service, Accra, Ghana

* akwadwo2003@yahoo.com

## Abstract

The moral implications of Assisted Reproductive Technology (ART) practices often pose a dilemma for those belonging to various cultural and religious groups. However, there is a gap in scientific studies examining the perspectives of spiritual leaders and their congregations regarding ART practices in Ghana. To bridge this gap, this paper employs a qualitative approach to investigate the religious, cultural, and personal interpretations of ART practices, including gamete donation, surrogacy, and cryopreservation, among 30 members of the key religions in Ghana. The study used in-depth interviews and focus group discussions to collect data, and the results were analysed thematically. The findings show that Islamic and Christian religious figures support the use of ART to treat infertility among married couples who use their gametes. However, they do not condone the use of third-party reproductive resources, such as gamete donation, surrogacy and cryopreservation. Traditionalists, on the other hand, do not endorse the use of ART in childbearing, as they believe it interferes with natural procreation processes and challenges the role of the Creator. Cultural, religious, ethical, and personal beliefs about procreation may have influenced the views on the acceptability of ART practices among the Traditionalists. It is, therefore, important to educate the public on theology, medical technology, and infertility while respecting individuals' religious beliefs and values.

## Introduction

In Ghana, approximately 13% of women of childbearing age experience infertility [1]. Historically, infertility has been associated with a range of supernatural and social explanations including witchcraft, envious neighbours, past relationships, ancestral or divine curses, and malevolent spirits [2,3,1]. In contrast, modern scientific inquiry has identified biological and lifestyle factors as the primary causes of infertility [4].

**Data availability statement:** All relevant data are within the article.

**Funding:** The author(s) received no specific funding for this work.

Previous studies have highlighted the negative impacts that childlessness can have on couples. Women who are unable to conceive often face physical abuse, social stigmatisation, economic hardship, and social segregation [5,6]. These negative impacts, societal pressure and the perceived value of parenthood, often motivate, individuals to explore wide range of solutions including seeking the guidance of religious leaders and utilising traditional beliefs and practices [7]. For instance, some couples who follow traditional faiths may use amulets and fertility symbols or consult fertility fetishes to increase their chances of conceiving. Others may rely on charms, traditional rituals, or drink special herbal preparations [8,9]. It is also common for couples with fertility challenges to consult Mallams and prayer camps for spiritual assistance, while others use religious artefacts such as necklaces, bracelets, stickers and oil for prayers with the hope of getting their infertility situation solved.

In recent times, couples facing fertility challenges often turn to conventional medical treatments such as Assisted Reproductive Technologies (ART). Couples often pursue these procedures after seeking guidance from religious leaders and seeking solace in their faith [10]. ART encompasses a range of treatments and procedures designed to help establish pregnancy through the handling of human oocytes, sperm, or embryos outside of the body [11]. These methods include in vitro fertilisation and embryo transfer, gamete intra-fallopian transfer, zygote intra-fallopian transfer, tubal embryo transfer, gamete and embryo cryopreservation, oocyte and embryo donation, and gestational surrogacy.

Despite increased awareness and availability of ART options such as gamete donation, cryopreservation, and surrogacy, societal and religious attitudes towards these practices as a means of achieving parenthood continue to pose a challenge for many couples. For instance, the concerns about the moral appropriateness and safety of such interventions, as well as cultural and religious beliefs surrounding procreation and motherhood, have discouraged individuals from seeking treatment [10,8].

Although ART practices and procedures have been widely discussed in religious circles, particularly in studies conducted by Hiadzi and colleagues [12], Fortier [13], Kooli [14], and Zachariah [15], few have explored the influence of culture and the perspectives of spiritual leaders and their congregations on these practices in Ghana. Against this backdrop, the present study aims to investigate the religious, cultural, and personal interpretations of key religious leaders and figures from Christianity, Islam, and Traditional religion in Ghana regarding the acceptability of ART procedures as a solution to infertility.

## Theoretical framework

The God-centric framework, often referred to as theocentrism, provides the theological lens for examining the moral reasoning and acceptance of ART practices within Traditionalist, Islamic, and Christian communities in Ghana. Central to this framework is the belief that God is the 'origin and sustainer of life', and that human reproduction is a sacred gift rather than a purely biological process. Consequently, any human intervention in reproduction, including the use of ART to address infertility, must be consistent with divine intent. The application of the God-centric framework

to Traditionalist, Islamic, and Christian interpretations of ART practices in Ghana reveals a nuanced interplay between spiritual belief systems and biomedical innovation. Across all three traditions, reproduction is not merely a biological event but a sacred process governed by divine authority, moral teachings and other factors.

The theocentric framework draws upon Bronfenbrenner's ecological theory to emphasise how a complex web of relationships and the environments, situated within a divine order, shapes individuals decision [16]. The ecological model broadens the analysis by examining how individual, interpersonal, community, institutional, and policy-level factors interact to shape the ART landscape in Ghana. At the individual level, personal beliefs and emotional experiences with infertility play a role in seeking treatment. Interpersonal dynamics, such as spousal support or family pressure, affect an individual's decision-making to seek ART treatment. Community norms and religious belief also influence the discourse about the acceptability of ART practices, while healthcare systems and government policies determine the availability, accessibility and regulation of ART services in Ghana. Combining these two models highlight a comprehensive understanding of ART in Ghana, which respects spiritual values, acknowledges cultural and economic contexts, and considers the broader socioecological system in reproduction decisions.

## Methods

### Ethics statement

The University of Cape Coast Institutional Review Board (UCC/IRB/A/2016/54) and the Ghana Health Service Ethics Review Committee (GHS-ERC: 02/10/2016) granted ethical approval for the study. Additionally, all participants provided formal written consent prior to their involvement in the study.

### Setting

Accra, the capital city of Ghana and a hub for the country's fertility hospitals was the study site. It has an estimated population of 2.27 million with approximately 83% of its inhabitants being Christians. Muslims make 11.8% of the population while 4.7% are Traditionalists [17].

### Design

The research aimed to explore the religious and cultural acceptability of different assisted reproductive techniques and practices in Ghana using a qualitative research approach. The study selected 30 participants through purposive sampling and utilised semi-structured in-depth interviews and focus group discussions to collect data.

### Sampling and data collection

The study recruited 30 individuals from Christian, Islamic, and Traditional religious backgrounds ensuring diverse perspectives on ART. The study adhered to the principle of data saturation to determine the sample size and data collection process. As outlined by Guest, Bunce, and Johnson [18], saturation is achieved when no new themes or insights emerge from additional interviews, indicating that sufficient data have been collected to support robust analysis. This principle was the benchmark in the determination of the sample size for the study.

The recruitment started on March 8, 2017, and ended on August 10 of the same year. The study purposely selected key religious leaders and other members within each faith group to participate. The Christian Group (14 participants) comprised 7 males and 7 females, including 5 religious leaders and 9 members. They represented Orthodox churches (Presbyterian, Methodist, Anglican), Pentecostal/Charismatic churches (Church of Pentecost, Assemblies of God, Lighthouse Chapel International), and Roman Catholic Church. The Islamic Group (10 participants) included a representative of the National Chief Imam of Ghana and 9 practising Muslims (4 males and 5 females). The Traditional Religion Group (6 participants) consisted of 3 males and 3 females.

An in-depth interview guide was utilised to gather insightful information from 6 significant religious figures, who were interviewed separately. The in-depth interview method was utilised to obtain detailed information regarding perspectives, experiences, feelings, beliefs, and the meanings assigned to ART by the selected religious figures. Due to the sensitivity of the topic, the in-depth interviews offered a safe space for key religious figures to express their perspectives freely to the principal investigator. Additionally, the investigator conducted five focus group discussions (FGDs) involving 24 people, 2 male groups and three female groups, each with either 7 or 5 participants. The participants represented a diverse mix of Christians, Muslims, and Traditionalists, ensuring a balanced exchange of perspectives.

All participants selected for the study were aged between 18 and 50 years and were all married, except for two Catholic Priests. Each participant has previous knowledge of, or experience with ART and identified with one of the three main religions. Participants came from diverse age groups, educational backgrounds, and residential statuses. The principal investigator conducted the IDIs and FGDs in quiet rooms at the study site while a trained research assistant documented the key points. Each session lasted for approximately 50 min. To maintain the integrity of the data, all discussions were recorded using digital means.

The interviewer assigned a unique code to each participant to maintain confidentiality and subsequently used these identifiers to organize and analyse the data. The research team structured the interview guides for both the IDIs and FGDs around key thematic areas. These included religious beliefs about the use of ART for infertility treatment, perceptions of practices like gamete donation, surrogacy, cryopreservation, and the acceptability of children born through ART. The principal investigator conducted the IDIs and the FGDs in both English and Twi, a local dialect spoken by the Akan people.

The principal investigator conducted all IDIs in English, while FGDs utilised both English and Twi to promote clarity and inclusivity. This bilingual approach enabled six participants who were not fluent in English to express themselves comfortably in their local dialect. Before audio recording began, participants gave informed consent. The principal investigator translated Twi responses using a back-translation method to ensure accuracy, and resolved any discrepancies by consulting field notes and interview recordings. To enhance the reliability of the data, he employed the peer review and member checking techniques.

The study ensured fair representation by including participants from Islamic, Christian, and Traditional religious backgrounds, promoting diverse perspectives on ART. Both males and females were involved to minimise gender bias, while five FGDs separated by gender helped balance power dynamics and encourage open discussions. Additionally, a mix of age groups, statuses, and experiences within the religious community ensured a broader and more representative sample.

### Data management and analysis

NVIVO version 12 was utilised to categorise the themes and sub-themes that arose during the analysis of respondents' interviews. The investigator assigned specific numbers to each transcribed participant's narrative to obtain the themes and categories. To ensure accurate coding, the transcripts underwent multiple readings to identify recurring patterns. This entailed meticulously examining and comparing expressions to determine how they aligned with the study's themes. The researcher systematically named consistent phenomena and assigned them to thematic categories that reflect the central ideas within the data. Table 1 presents the related themes organised into clusters for in-depth thematic analysis.

### Results

#### Respondents' background characteristics

Table 2 presents the respondents' characteristics. From Table 2, half of the respondents (15) were aged 30–39 years, 12 had a secondary education, and 26 were married. In addition, 14 respondents were Christians, 10 were Muslims, and 6 belonged to the Traditional region.

**Table 1. Themes and categories obtained from participants on the acceptability of ART and practices.**

| Themes | Sub-themes |
|---|---|
| **Treatment of infertility** | |
| Muslims | Seeking the Face of Allah |
| | Use of talisman |
| | Washing the affected part of the body |
| | Fertility and Infertility must be accepted as coming from God |
| Christians | Seeking the Face of God |
| | Childbearing is a gift from God |
| | Infertility must be accepted after prayers |
| Traditionalists | Use of traditional/herbal medicines |
| | Use of prayers, traditional and ART |
| **Acceptance of ART as a treatment for infertility** | |
| Muslims | Accept treatment with spouses' reproductive resources (sperm and egg) |
| Christians | Accept treatment with spouses' reproductive resources (sperm and egg) |
| Traditionalists | Do not believe in ART treatment |
| | Inappropriate |
| | Man could be likened to God |
| **Acceptance of donor gamete** | |
| Muslims | Do not accept donor gamete because of genetic and spiritual implications |
| | Soiled a woman |
| | Adultery |
| Christians | Disapprove because it has ethical, social and spiritual consequences |
| | Adultery |
| Traditionalists | Disapprove because it is against the will of God |
| | Punishable by God |
| | Adultery |
| **Acceptance of surrogacy** | |
| Muslims | Disallow because of ethical reasons |
| | This may lead to incest |
| Christians | Disapprove because it is not biblical |
| | Creates no bond |
| Traditionalists | Disapprove because it is not natural |
| | No bond between mothers and children |
| **Acceptance of cryopreservation** | |
| Muslims | Accept cryopreservation of spouse's gametes |
| Christians | Accept cryopreservation of spouse's gametes |
| Traditionalists | Disapprove/inappropriate/impossible |
| **Perception of children born through ART** | |
| Muslims | A biological child if born with spouse's gamete |
| | Born out of wedlock if born with donor reproductive resources |
| | Cannot inherit the property of the adopted father |
| Christians | A biological child |
| Traditionalists | Born out of wedlock |
| | Children may have development challenges |
| | Children may grow and have health challenges |

**Table 2. Socio-demographic profile of participants.**

| Respondents | Number |
|---|---|
| **Gender** | |
| Males | 14 |
| Female | 16 |
| **Agegroup** | |
| 20-29 | 7 |
| 30-39 | 15 |
| 40+ | 8 |
| **Education** | |
| Noformaleducation | 4 |
| JHS/Middle | 8 |
| Secondary | 12 |
| Tertiary | 6 |
| **Maritalstatus** | |
| Married | 26 |
| Notmarried | 4 |
| **Religion** | |
| Christianity | 14 |
| Muslim | 10 |
| Traditionalist | 6 |

## Treatment options for infertility among religious groups

In most cases, people who have fertility challenges seek various treatment choices, including the use of traditional medications and religious assistance. Both Christian and Islamic leaders believed that infertility treatment was through prayers. A Catholic Priest had this to say:

> "We Christians seriously believe that God gives children, and so we always pray and counsel our members who have fertility challenges to seek the face of the Lord. The use of any unnatural means to have children, such as the use of ART, is not accepted".
>
> (A Catholic Priest, 45 years old, IDI)

Just as Christians, the participants from the Islamic religion were emphatic on the following infertility treatment options based on the Islamic faith:

> "The use of prayers which 'blows off' ailment of any part of the body that is affected is a means of treating conditions including infertility".
>
> (A Muslim father, 40 years old, FGD participant)

Contrary to the observations made from the Christian and Islamic religions, the participant from the Traditional religion noted that:

> 'Infertility is best treated with the use of traditional herbal medications and children born through this means are strong and healthy".

(A Traditionalist mother, 47 years old, FGD participant)

Insights from the FGDs provided further context to the various treatment options for infertility. It emerged that the combination of prayers and medications (orthodox, traditional, or the combination of the two) could address infertility challenges as typified in the excerpt below:

"Prayers without action cannot solve your fertility challenge. I am not saying that prayer at this time is not good, but you need to back your prayers with actions such as taking medications. After all, God works through man".

(A Christian mother, 28 years old, Anglican Church, FGD participant).

### The general perception of the use of ART as a treatment option

Apart from participants from the Roman Catholic Church who abhorred ART use as an infertility treatment option, those from the other religious groups appeared not to have any issue with the use of ART as an infertility treatment option, especially among married couples. From the Islamic view, the representative of the Chief Imam remarked:

'According to the Holy Qur'an, 23:5, Islamic laws do not frown on attempts to cure infertility among Muslims. In a situation where a man's sperm is put together with the wife's egg by a doctor to assist them to have children, Islam permits it".

(An Imam, 48 years old, IDI)

The perception of the Traditionalists regarding ART use for infertility treatment was doubtful:

"I have heard that sperm and eggs could be collected and a child created in a bottle for couples. To me, I doubt if that is possible...Can you create a fellow man. Are you God?"

(A Traditionalist mother, 33 years old, FGD participant)

Key religious figures from both the Orthodox and Pentecostal/Charismatic churches also believe that there is nothing wrong if couples use their reproductive resources (sperm and egg) to have children through the application of ART techniques. For example, a religious leader from the Presbyterian Church of Ghana made these observations on the subject:

"The use of ART by needy couples is accepted, provided the egg and sperm come from the couples themselves. This is because Christians believe that physical intimacy between a husband and wife remains a Biblical means of producing children".

(A religious figure, 38 years old, Presbyterian Church of Ghana, IDI)

### Perspectives on gametes donation and use

Gamete donation explains a situation where an individual offers him or herself to give out sperm or oocytes (eggs) to prospective individuals or couples whose situation calls for the use of sperm or eggs to undergo fertility treatment. Below are excerpts from the interactions with a leader from the Church of Pentecost (COP):

'The Church does not encourage its male members to donate or sell their sperm to needy couples. In the same way, the church prohibits gametes from a fellow woman for pregnancy purposes. Often, there are serious social, ethical, psychological and even spiritual consequences that emanate from using these reproduction resources'

(A religious figure, 37 years old, Church of Pentecost, IDI)

Elucidated below is the position of Islamic participants about the use of donor gamete:

"Islam does not allow gamete donation. It is believed that there is something genetic and spiritual about donated gametes and therefore it is not allowed".

(An Imam, 48 years old, IDI)

The FGD session with the men's group confirmed the views shared by the religious leaders. They unanimously condemned gamete donation as a treatment option for infertility. A male participant noted that:

"I do not have any challenge with children produced by the use of a couple's reproductive products (sperm and egg). My difficulty has to do with another man or woman donating their gametes. This act is akin to fornication, and God frowns on that. I cannot take another man's sperm for my wife to produce a baby for me. How would I call this child? For God's sake, let us not encourage these practices in the country"

(A Christian father, 41 years old, Methodist Church, FGD participant)

In another FGD held with a women's group, they expressed mixed reactions to the use of donor gametes to meet reproductive desires. In elaborating on the issues, a woman retorted:

"For me, as a woman, I will accept gamete donation if it is the only means that will help me to have a child. After all, society blames women when couples face fertility challenges in marriages, and if my church is against this act, I will simply stop going to that church so that I can have my children through the use of donor services".

(A Christian mother, 26 years old, Methodist Church, FGD 3 participant)

**Surrogacy**

Surrogacy implies that a woman becomes pregnant and gives birth to a child to give the child away to another person or couple after delivery. The participant from the Church of Pentecost (COP) had this to say about surrogacy services:

"The Church discourages its members from accessing surrogacy services or third-party methods in the ART treatment. Female members of the Church are also discouraged from offering themselves as surrogate mothers because it is against the will of God".

(A religious leader, 46 years old, Church of Pentecost, IDI participant)

On the same issue, participants from the Islamic and the Traditional religions submit that:

"Islam does not accept surrogacy. Instead, Islam allows adoption, but you must identify the biological parents of the adopted child. Islam says we should preserve the identity of an adopted child concerning his/her genealogy. Anything less than this is a sin against God".

(An Imam, 48 years old, IDI participant)

"Will you see such a child be your blood when you did not carry it yourself? God knows why He puts babies into the womb for nine months. It is to create a bond between the mother and the child. If you practice surrogacy, the child will grow and may not have any regard for the parents because the mother did not carry the pregnancy".

(A 27-year-old Traditionalist, FGD participant)

### Cryopreservation

Fertility cryopreservation is the conservation of sperm, oocytes, embryos, and other reproductive tissues to help in reproduction. The various religious groups in Ghana have not wholly accepted this practice. For example, the interviewee from the COP stated that:

> 'Cryopreservation is not accepted by the Church. Alternatively, whenever freezing becomes necessary, freeze the wife's eggs and the husband's sperm separately and be used by the same couple while the couple is alive.
>
> (A religious figure, 37 years old Church of Pentecost, IDI participant)

Likewise, the Islamic cleric concurred with the Christian participants' view on cryopreservation and provided further insights on the use of Cryopreservation explaining that:

> "Because death and divorce are possible, cryopreservation is not allowed. In Islamic laws, once there is a divorce or a spouse dies, cryopreserved sperm or egg cannot be used by the surviving spouse".
>
> (An Imam, 48 years old, IDI participant)

In the FGDs, men said there was no need for cryopreservation because they could give birth at any age; they thought that cryopreservation could render the gametes impotent. A discussant during the FGD made these revelations:

> "What is the purpose of freezing a person's eggs or sperm? I do not think this is possible. You see, when you keep meat or any food item in the fridge for a long time, the taste changes. In the same way, I believe the gametes may not be functional after keeping them in the fridge for a long time". (A Christian father, 33 years old, Assemblies of God Church, FGD participant)

### Perception of children born with ART

Despite the increased usage of ART and the total number of children born through technology, many still believe that children born using ART may not be healthy due to human handling of the gametes. On this issue, the Traditionalists questioned the ability of couples to bear healthy children using ART:

> "I have not seen any child born through the use of ART but I doubt whether they will be normal. I am sure the children we see around who are handicapped and present other physical challenges may be as a result of the use of this new procedure".
>
> (A Traditionalist father, 36 years old, FGD participant)

Contrary to the above, views expressed by the Christian groups were however at variance with the perception held by the Traditionalists. They noted that the church accepts children born using ART. In explaining this issue, this was what a church leader from Light House International Chapel had to say:

> "Children born through ART services are human beings like you and I. We dedicate them to God as long as they come from the sperm and egg of a man. You can not show any form of discrimination to such children".
>
> (A religious figure, 39 years old, Light House International Chapel, IDI participant)

The views shared by leaders of the Islamic faith, though somewhat like those of Christian participants, significant variations were recognised; especially on the conditions under which children born with ART were acceptable. In buttressing the position of the Islamic faith, this was how an Imam put it:

"Children born through ART services are legitimate, provided that the couples used their gametes. On the other hand, if the child is born with donor reproductive resources, it will be accepted as a human being, but that child will not inherit the property of the adopted father".

(An Imam, 48 years old, IDI participant)

The female participants echoed the Christians stance on children born through ART during an FGD session. They argued that children born using ART are just like those born through natural means. The views shared during deliberations with them are summarised in this excerpt by one of them, indicating that.

"Children born through the use of ART are also human beings, strong and beautiful like every normal being. If not told, you will never know that such children were born with ART. I have seen some of these children, and there is nothing wrong with them".

(A Christian mother, 27 years old, Roman Catholic Church, FGD participant)

## Discussion

The study revealed that Muslims, Christians and Traditionalists recognised the significance of marriage, fertility and procreation, values deeply rooted in the belief that lineage and family continuity are crucial for community stability. However, religious and cultural beliefs significantly influenced infertility treatments, with key religious leaders and members expressing varied and subjective stances. The Traditionalists in particular, favoured the use of herbal remedies and concoctions over biomedical approaches like ART for the treatment of infertility. Previous studies have confirmed the use of herbal and complementary medicines (CAM) in addressing infertility. For instance, Nazim and colleagues [9] reported that herbal practitioners in northwest Algeria used a range of remedies to treat primary infertility in females. Similarly, Kuug and colleagues [19] discovered that communities in the Nandom and Talensi districts of northern Ghana preferred traditional remedies once the cause of infertility was traditional. This is consistent with findings from rural Zimbabwe, where infertility treatment was exclusively traditional. Factors such as accessibility, affordability, alignment with local beliefs and religious values and the belief in natural remedies may have contributed to the widespread use of traditional and complementary medicine. Nonetheless, concerns and doubts persist regarding the safety and effectiveness of these treatments [20].

Religious leaders within both Islamic and Christian communities emphasised that seeking God through continuous fasting, prayer, and religious activities could lead to healing from all diseases, including infertility. Christian participants regarded motherhood as a divine gift that was sacred to humanity. Couples, therefore, prioritised religious activities as the primary solution for infertility. As Stern (2018) noted, God knew the situations of Abraham and Sarah (Genesis 15:1–5), Jacob and Rachel (Genesis 29:31), Zechariah and Elizabeth (Luke1:5–7) and in His own time answered their prayers and gave them children. Despite this stance, other participants from both religious groups believed that the combination of traditional, orthodox medicine and prayer (religion and medicine) was the best approach to treating infertility. The integration of religion and medicine have been observed to be the common coping strategies people use to address life challenges, including infertility [21]. In most situations, societal and familial pressure to achieve parenthood shortly after marriage influence the decision to seek complementary treatments.

Islamic clerics affirm that ART is permissible under certain conditions. For instance, Muslim jurists, including both Sunni and Shia scholars, generally agree on the significance of fertility, procreation and the pursuit of infertility treatments. They share a common understanding of the value of procreation within the confines of marriage. However, there are important differences regarding specific practices within modern ART. Key areas of divergence include views on third-party donations (sperm, egg and embryo), surrogacy and sex selection.

Sunni scholars consistently reject the use of third-party reproductive resources, including sperm, egg, and embryo donations due to religious concerns about lineage and the importance of clear parental lines [22]. For Sunnis, maintaining the integrity of family connections is paramount, and as a result, some may choose to remain childless rather than violate their religious prohibitions against ART practices. Studies have also shown that within Sunni communities including countries like Turkey, Jordan and Egypt, the use of third-party gametes is linked to zina (adultery), reflecting strong religious restrictions to donor involvement in reproduction [23]. Additionally, there have been concerns about long-term social implications about the use of donor gamete. This include potential risk of unintentional incest, particularly if two individuals conceived through anonymous donation unknowingly enter into a marital relationship later in life [24].

However, there have been instances where parents within the Sunni community preferred having non-biological children to being childless entirely [25,26]. Other individuals pursued donor sperm treatments without informing their spouses. For example, some wives have collaborated with medical practitioners to use donor sperm without the knowledge of their husbands, and in some situations, husbands have done the same without the consent of their wives [25,27] to meet their parenthood desires. These actions, often described as spousal deception, motivated by the desire to meet parenthood aspirations while avoiding social stigma associated with infertility or the perception of being spiritually inadequate in the context of marriage. Couples may embark on fertility treatments involving third-party reproductive resources to preserve family integrity and continuity [27], maintain marital harmony and to avoid familial and sociocultural pressure due to infertility [27]. When such methods result in successful conception, many couples choose to keep the involvement of third-party donors confidential from extended family members and the children born through these means for the fear of stigma.

In contrast, many Shia scholars adopt a more permissive stance regarding third-party reproductive resources. For example, in Shia countries like Iran and Lebanon, practices such as embryo donation and surrogacy are often deemed permissible, even in cases of temporary marriage (mut'a), viewing these methods as consistent with Islamic legal and religious principles [28].

Accordingly, Christian Orthodox churches allow the use of ART among their members who are couples [29]. Kukin [30] views medical advancement and technology as blessings from God. He argues that using pharmaceutical treatment or various surgeries to enhance pregnancy does not disrupt intimacy between partners; rather, it supports and restores the natural function of the body. However, there are those, such as the Vatican and Traditionalists, who view ART as immoral and contrary to biblical teachings. Some Traditionalists even question whether ART can truly fulfil the desire for parenthood, while others argue that its use equates to man playing God, which they deem unacceptable.

The study also found that leaders from Traditional, Islamic and Christian backgrounds unanimously opposed the use of donor gametes, cryopreservation and surrogacy. They assume that the use of donor gametes constitutes a third party's interference in a sanctified marital union and may be perceived as adultery, an act considered punishable by God and man-made laws. Consequently, in Ghanaian society, these actions are incompatible with moral and cultural values. This finding is consistent with a study that established that many women considered gamete donation as adultery [31]. Another study in France observed that women who received sperm from men other than their legal husbands felt 'uncleaned'. They also perceived that surrogacy does not create a bond between parents and their children because the birth of the child is from third-party reproductive resources [32].

In some situations, partners may not show the desired love and affection for the children who are born with donor gametes, while the children may do the same when they realise that they were born with donor gametes. This perspective is based on the premise that the child was not born through the intimate union of the couple but was medically made thereby challenging the traditional notion about procreation [33]. Other studies [34,35,29] have concluded that ART practices such as donor gametes, surrogate motherhood and cloning are morally unacceptable. This finding corroborates a statement by Francis [36] "the path to peace calls for respect for life, for every human life, starting with the life of the unborn child in the mother's womb, which cannot be suppressed or turned into an object of trafficking. In this regard, the practice of the so-called surrogate motherhood represents a grave violation of the dignity of the woman and the child, based on the

exploitation of situations of the mother's material needs". He therefore called for the global ban of surrogacy referring to it as exploitative and morally unacceptable [36].

In Ghanaian society, where donor gametes and surrogacy are not popular, this may raise queries about the meaning of the family and family privacy as well as the welfare of the child when the use of donor gametes and surrogacy becomes public. Again, the perception that donor gamete and surrogacy could also lead to incest in the future, a view held by the Sunni Muslims [31] persist in Ghana. Previous studies have also outlined several factors as the consequences of using ART. For example, Valji [37] outlined three reasons why Christians must reject the use of third-party gametes. First, the perception that utilisation of ART violates the sanctity of marriage; second, the belief that a third-party gamete in the reproduction processes leads to the exploitation of human beings; and third, the belief that ART does not command respect for the rights of children born through the act. The Roman Catholic Church has cited two major reasons why its members should shun the use of third-party reproductive resources. First, the Church believes that a human embryo is a soul. Second, there is the belief that procreation must only occur within the confines of sexual intercourse within marriage and between a male and a female [38,39].

Traditionalists believe that using donor gametes and surrogacy can alter the genetic identity of children born through these processes. They further argue that the use of donor gametes should be discouraged because the impregnation of a wife by a third-party is considered a violation of marital sanctity, an act punishable by both divine and man-made laws. This notion reflects a male-centric control over female sexuality and reproduction and positions the husband as the sole legitimate source of reproduction, reinforcing patriarchal authority over the woman's body. Moreover, the belief that a true mother is the one who successfully goes through pregnancy and spontaneous vaginal delivery (SVD) influences the decision not to opt for surrogacy and other ART procedures involving a third-party's reproductive resource. Women often see SVD as a natural way to demonstrate their competence and ability in maternal roles and responsibilities. For example, women from northern Ghana prefer SVD as it is perceived as natural and offers a quicker recovery [40], while other women believe that it creates a bonding relationship between mothers and children [41]. It also reinforces the traditional gender role perception, where women are expected to bear children as a fulfilment of both their spiritual duty and marital obligations [42].

Islamic Sharia laws also prohibit the use of donor sperm and other third-party resources for fertilisation processes. It is believed that the use of donor gametes may result in a child whose biological parents may be different from conjugal parents and this may have consequences on legitimate conjugal relationships The practice of anonymous donation may also present the possibility of future incest between half-siblings, who may marry without knowing that they share biological parents [43]. Further, Sunni jurisprudence perceive surrogacy as inappropriate due to the potential confusion about the identity of the biological mother, which disrupts established family ties and lineage [44]. On the other hand, Shia scholars are generally more liberal regarding surrogacy arrangements, allowing them under certain conditions. They emphasise the circumstances and intentions behind such practices rather than an outright ban [45].

The study further revealed that Muslims, Christians and Traditionalists rejected the use of cryopreserved gametes by individuals lacking conjugal connections. However, Christians and Muslims explain that legally married couple could only use their cryopreserved gametes when they are alive and are still married. To prevent unauthorised use, ethical and legal guidelines require the disposal of stored gametes following the death of a spouse or the dissolution of a marriage.

Among Sunni Muslims, couples may opt to preserve surplus embryos; however, their use is strictly limited to the marital relationship, thereby reinforcing the moral boundaries of the family structure [46]. While Shia scholars also recognise the option of embryo preservation, their guidelines may allow for a wider interpretation of family dynamics, reflecting their more accommodating stance on reproductive practices.

Some other couples and individuals may defy religious and cultural beliefs and utilise ART to meet their parenthood desires [47]. For example, some studies have observed that couples and individuals have adopted secrecy to maintain privacy in accessing ART practices, which are observed to be religiously and culturally inappropriate [22,27,48]. Other

familial and sociocultural pressures to bear children have also motivated couples to defy religious and cultural beliefs surrounding the use of ART [27]. Again, couples may travel outside their home countries where certain ART procedures are restricted to have the preferred procedure done for them [49,50].

According to the study, people's perception of children born with ART varies across different religions. Traditionalists hold the belief that these children may have birth defects, developmental and growth challenges. Their perspective is shaped by the perception that humans manipulated the gametes leading to pregnancy and delivery. Per the argument of Opuku and Addai-Mensah [51], the use of ART practices and techniques, including donor gametes, surrogacy, and cryopreservation is "an intrusion in the divine process of procreation, an intrusion into the bond of marriage and parenthood, the sanctity of life and the status of the embryo."

By contrast, key religious figures from both the Islamic and Christian faiths perceive children born using ART as the creation of God, though they do not support a third-party's involvement in procreation. Christians believe that once a child is born, it does not matter whether it was born using third-party reproductive resources or not. According to their faith, the child is God's image and, therefore, must be dedicated to God. According to Anderson and Walker [52], regardless of how a child is born, it embodies 'Imago Dei' and assured of God's love and the love of the Christian community. On the other hand, Islamic principles accept such children as adopted children and are not entitled to inherit the father's property.

Despite the religious and cultural views surrounding children born through ART practices in Ghana, there is currently no law protecting the rights of these children. This situation contrasts sharply with that in some advanced countries where the current ethical debates focus on the modern ART methods, (surrogacy, embryo donation, embryo selections), public funding, access and policies [53,54]. Consequently, there is a pressing need for legal, sociocultural, ethical, and biological protections for these children. Despite efforts to address this issue, such as the European Parliament urging European countries to uphold the International Convention for the Protection of Human Rights, concerns remain. Children born through surrogacy have the same rights as all other children under the United Nations Convention on the Rights of the Child (CRC). Despite the existence of international frameworks, many nations have not accepted this concept. Legal and ethical recognition of children conceived through third-party reproductive methods continues to vary widely across jurisdictions. For example, Marinelli [55] argued that ART practices, such as surrogacy and gamete donation violate the rights of children. Yarema [56], highlights that surrogacy processes serve the interests of adults rather than the welfare of children, as they undergo various manipulations and lack the right to be conceived and born to their biological parents [56].

Additionally, the citizenship of children born through third-party resources and surrogacy presents challenges, especially when intended parents abandon them. In some countries, authorities place abandoned children in orphanages to ensure their care and shelter [57]. There is also an increasing number of reported court cases in Europe involving the use of gametes by former spouses, along with disputes regarding the custody of children born to surrogates [58,59].

## Limitation

The study focused on selected religious denominations within the Greater Accra region of Ghana, and the present findings may not be generalisable to the entire country. A future study should draw on nationally representative data to enhance the validity and generalizability of findings

## Implication of the study

The study highlights the need for a multifaceted approach to reproductive health, emphasising policy-driven education that addresses religious and ethical concerns while promoting informed decision-making in ART uptake. It underscores the importance of culturally sensitive healthcare services that respect religious values, norms and encourages collaboration between religious leaders, healthcare providers, and the public to reduce stigma and misinformation surrounding ART. Additionally, the findings reinforce ethical and regulatory frameworks that align medical advancements with societal values while safeguarding reproductive rights of the public.

## Conclusions

Religion and belief systems have historically played a significant role in shaping the acceptance or rejection of ART practices, such as donor gametes, surrogacy, and cryopreservation. In Ghana, religious leaders from the three main faiths assert that fertility and childbearing are divine gifts from God. As a result, couples facing infertility are encouraged to seek divine intervention through prayer or consider adoption. Practices like donor gametes and surrogacy, which involve reproductive resources from third-parties are generally viewed as unacceptable whiles Traditionalists do not support the idea of using ART for reproduction at all.

Despite the differing perspectives among Traditionalist, Islamic, and Christian groups regarding ART practices, there is a consensus that children born through these methods are precious gifts from God. This recognition applies not only to cases where third-party reproductive resources are used for conception but also stems from a shared belief in the sanctity of life and the importance of parental rights.

The perspectives held in Ghana reflect a broader global debate on ART and religion, where differing beliefs significantly influence societal acceptance of these technologies. In various cultures and religious communities around the world, conflicts often emerge between traditional beliefs and modern scientific practices. This ongoing discourse raises important questions about ethical considerations, the definition of family, and the role of technology in human reproduction. It is therefore crucial to integrate religious perspectives into campaign design, public health efforts to foster trust, reduce stigma, and improve ART uptake and adherence across diverse populations.

This calls for a study to understand the community's perceptions of using donor gametes, surrogacy, cryopreservation, and other third-party reproductive resources in Ghana. In addition, a study on the perspectives from young unmarried adults, who may represent the changing generation could be carried out. Insights gained from the studies could contribute to the global dialogue on ART and religion, highlighting the need for respectful engagement between scientific advancements and deeply held cultural beliefs.

It is important to note that, despite the data for the study (March - August 2017), religious attitudes towards ART in Ghana appear to have remained relatively stable. Both Christian and Islamic perspectives continue to support ART primarily within the bounds of marriage, whereas Traditionalist beliefs generally reflect disbelief.

### Reflexivity statement—insider Christian perspective

Growing up in a Christian home in Ghana has shaped my understanding of how faith influences views on reproduction. This personal background sparked my interest in how religion connects with science and personal choice, especially when it comes to Assisted Reproductive Technologies (ART). I am aware that my Christian beliefs may affect how I see these issues, so I have made a conscious effort to include other views by speaking with Islamic scholars and traditionalist practitioners. This approach has helped me to stay objective and respectful of all perspectives.

### Supporting information

**S1 Text. Study Instrument.**
(DOCX)

### Acknowledgments

The author expresses gratitude for the valuable contributions of all participants who shared their experiences on the topic. Additionally, the author recognises the helpful proofreading assistance of Ebenezer Agbaglo from the Department of English at the University of Cape Coast and is thankful for the support of research assistant, Mr. Francis Nkansah of the Ghana Health Service.

## Author contributions

**Conceptualization:** Kwadwo Asante-Afari.

**Data curation:** Kwadwo Asante-Afari.

**Formal analysis:** Kwadwo Asante-Afari.

**Investigation:** Kwadwo Asante-Afari.

**Methodology:** Kwadwo Asante-Afari.

**Resources:** Kwadwo Asante-Afari.

**Validation:** Kwadwo Asante-Afari.

**Visualization:** Kwadwo Asante-Afari.

**Writing – original draft:** Kwadwo Asante-Afari.

**Writing – review & editing:** Kwadwo Asante-Afari.

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
