## [Decision Letter · Decision Letter 0]

4 Sep 2024

PGPH-D-24-01257

Exploring the Intersection of Religion and the Acceptance of Assisted Reproductive Technology Practices: An Examination of Selected Religious Groups in Ghana

Dear Dr. Asante-Afari,

Thank you for submitting your manuscript to PLOS Global Public Health. After careful consideration, we feel that it has merit but does not fully meet PLOS Global Public Health’s publication criteria as it currently stands. Therefore, we invite you to submit a revised version of the manuscript that addresses the points raised during the review process.

The manuscript has been evaluated by two reviewers, and their comments are available below.

The reviewers have raised a number of concerns. Specifically, they recommend that you provide further details regarding the sample and data collection.

Could you please carefully revise the manuscript to address all comments raised?  I have also noticed that one reviewer has recommended that you cite specific previously published works. As always, I recommend that you please review and evaluate the requested works to determine whether they are relevant and should be cited. It is not a requirement to cite these works. 

We look forward to receiving your revised manuscript.

Kind regards,

Johanna Pruller, Ph.D.

PLOS Staff Editor

Journal Requirements:

Additional Editor Comments (if provided):

Reviewers' comments:

Reviewer's Responses to Questions

**Comments to the Author**

1. Does this manuscript meet PLOS Global Public Health’s publication criteria?

Reviewer #1: Yes

Reviewer #2: Yes

2. Has the statistical analysis been performed appropriately and rigorously?

Reviewer #1: Yes

Reviewer #2: Yes

3. Have the authors made all data underlying the findings in their manuscript fully available (please refer to the Data Availability Statement at the start of the manuscript PDF file)?

Reviewer #1: Yes

Reviewer #2: Yes

4. Is the manuscript presented in an intelligible fashion and written in standard English?

Reviewer #1: Yes

Reviewer #2: Yes

Reviewer #1: Dear authors,

I read with great interest the manuscript, which falls within the aim of this Journal. In my honest opinion, the topic is interesting enough to attract the readers’ attention. Nevertheless, authors should clarify some points and improve the discussion, as suggested below. Authors should consider the following recommendations:

In my opinion you have to improve the paper refering in the text to the updated literature on this topic focusing how its really important HOW IS IMPORTANT to preserv their fertility before a fertility sparing treatment for future use of their oocyte ESPECIALLY IN CASE OF MALIGNANCIES DIAGNOSIS and to perform a NIPT test at beginning of pregnancy and to follow th neonatal Outcomes and Long-Term Follow-Up of Children Born from Frozen Embryo.

I suggest you to read and cite these articles:

Fertility-Sparing Approach in Patients with Endometrioid Endometrial Cancer Grade 2 Stage IA (FIGO): A Qualitative Systematic Review

Endometrial Cancer in Reproductive Age: Fertility-Sparing Approach and Reproductive Outcomes

Fertility-Sparing Strategies for Early-Stage Endometrial Cancer: Stepping towards Precision Medicine Based on the Molecular Fingerprint

Impact of assisted reproduction techniques on the neuro-psycho-motor outcome of newborns: a critical appraisal

Sentinel lymph node staging in early-stage cervical cancer: a systematic review

Neonatal Outcomes and Long-Term Follow-Up of Children Born from Frozen Embryo, a Narrative Review of Latest Research Findings

Circulating miRNAs as a Tool for Early Diagnosis of Endometrial Cancer-Implications for the Fertility-Sparing Process: Clinical, Biological, and Legal Aspects

Measuring the composition of the tumor microenvironment with transcriptome analysis: past, present and future

Cell-Free Fetal DNA and Non-Invasive Prenatal Diagnosis of Chromosomopathies and Pediatric Monogenic Diseases: A Critical Appraisal and Medicolegal Remarks

Assisted Reproductive Techniques and Risk of Congenital Heart Diseases in Children: a Systematic Review and Meta-analysis

Fresh vs. frozen embryo transfer in assisted reproductive techniques: a single center retrospective cohort study and ethical-legal implications

Stem Cells and Infertility: A Review of Clinical Applications and Legal Frameworks

Reviewer #2: Dear Author

First of all I would like to congratulate you for this careful and interesting work. It is a very "Ghanaian" look at aspects of infertility and assisted reproduction, but at the same time we can extrapolate some of these feelings and testimonies as universal.

Acronyms

The paper is well written, only one observation is related to the acronyms, which should be next to the first reference, in parentheses. This does not occur, for example, in Design, where semi-stuctured in depth interviews and focus group discussions are not followed by IDI and FGDs. Next, when reading Sampling and data collection, we have to go back to understand the acronyms.

In the fourth line of discussion, the acronym CAM is used for complementary medicine. However, this reference is not repeated, and is probably dispensable.

Sampling and data collection

The author states that Christian individuals (Orthodox, Protestant and Catholic) were heard, as well as Muslims and members of the Traditional African religion, religious figures and practising members of the communities.

Later Methodists, Anglicans, and Presbyterians are cited as Orthodox, others distinct as Pentecostals, Assemblies of God Church, and Light House Chapel. This nomination as Orthodox seems to me not adequate. Perhaps better consider them all as Christians, yes, but Protestants. In fact the last three in my country (Brazil) probably being more radical.

Traditional- was the group that caught my attention, for an increasingly radical conduct. Would they be older or with a lower degree of formal education?

Religious leaders tended to position themselves more vehemently against the procedures of gamete donation, cryopreservation and surrogacy. It is interesting their citing of spiritual consequences, while 2 practicing Methodists would consider either as adultery (man), or the woman leaving the church to undergo treatment.

The Traditional group again intrigued me by attributing to assisted reproduction the desire of man to compare himself to God, going against God's will, adultery etc. Sometimes it seems to me more like a position of other religions in their radical branches. Wasn't traditional herbal medicine associated with amulets, offerings, or the like?

And, was there no condition of religious syncretism reported in these testimonies of practitioners? I expected to find a traditionalist mention of various entities of gods, not a single reference.

Finally, the testimonies collected among the practitioners demonstrate a culturally evolving Ghanaian society, where concepts related to family foundation can be better defined.

Regards.

**Do you want your identity to be public for this peer review?** For information about this choice, including consent withdrawal, please see our Privacy Policy

Reviewer #1: **Yes: ** GIUSEPPE GULLO

Reviewer #2: **Yes: ** Maria do Carmo Borges de Souza

---

## [Author Response · Author response to Decision Letter 1]

1 Oct 2024

September 23, 2024

Dear Editor

Response to Reviewers' comments

Thank you for sending the reviewers’ comments on the manuscript "Exploring the Intersection of Religion and the Acceptance of Assisted Reproductive Technology Practices: An Examination of Selected Religious Groups in Ghana" to me for the opportunity to address them for your consideration. I have addressed all the comments and provided some responses and clarifications where necessary as per the attached. I have also attached a track-changed and untracked version for your review. I believe the manuscript has improved vastly and look forward to your kind consideration for acceptance and publication.

Thank you.

Kwadwo Asante-Afari

(Correspondent)

Reviewer(s)' Comments to Author:

Acronyms

1. Comment: In-depth interviews and Focus Group Discussions were not followed by acronyms. When reading Sampling and data collection, we have to go back to understand the acronyms.

Response: Acronyms for In-Depth Interviews (IDIs) and Focus Group Discussions (FGDs) are defined upon first use and used consistently throughout the study. See pages 5-7 for details.

2. Comment: The author states that Christian individuals (Orthodox, Protestant and Catholic) were heard, as well as Muslims and members of the Traditional African religion, religious figures and practising members of the communities.

Later Methodists, Anglicans, and Presbyterians are cited as Orthodox, others distinct as Pentecostals, Assemblies of God Church, and Light House Chapel. This nomination as Orthodox seems to me not adequate. Perhaps better to consider them all as Christians, yes, but Protestants. In fact, the last three in my country (Brazil) probably being more radical.

Response: Referring to Methodists, Anglicans, and Presbyterians as Orthodox has been changed to Protestants as suggested by the reviewer. Please see paragraph 1 of page 6 for details

3. Comment: In the fourth line of discussion, the acronym CAM is used for complementary medicine. However, this reference is not repeated and is probably dispensable.

Response: The use of the CAM acronym has been discontinued because of its misplaced application.

Comments: Traditional- was the group that caught my attention, for an increasingly radical conduct. Would they be older or with a lower degree of formal education?

Response: Most of the respondents who belonged to the Traditional African religion were old (40+ years) and had a lower degree of formal education. See the first paragraph under ‘Respondent background characteristics’ on page 10 for further details

Comment

The Traditional group again intrigued me by attributing to assisted reproduction the desire of man to compare himself to God, going against God's will, adultery etc. Sometimes it seems to me more like a position of other religions in their radical branches. Wasn't traditional herbal medicine associated with amulets, offerings, or the like?

Response: Yes traditional herbal medicine was associated with amulets, offerings, or the like but also related to the taking of herbal medication, concoctions, the wearing of artefacts and many more. Please see pages 11 and 18 (first paragraph) under discussion for information on this comment.

Comment

And, was there no condition of religious syncretism reported in these testimonies of practitioners? I expected to find a traditionalist mention of various entities of gods, but not a single reference

Response: there was a condition of religious syncretism reported in these testimonies of practitioners. For example, on page 18, participants from both Christian and Islamic backgrounds reported combining traditional and orthodox medicine with prayer (religion and medicine) to address infertility and this is a clear example of religious syncretism. This has been explained in the first paragraph of page 18 (Discussions)

Response References have been duly revised and crosschecked to ensure that all references cited in intext have been addressed in the main reference column. Please see the reference section for further details

---

## [Decision Letter · Decision Letter 1]

16 Jan 2025

PGPH-D-24-01257R1

Exploring the Intersection of Religion and the Acceptance of Assisted Reproductive Technology Practices: An Examination of Selected Religious Groups in Ghana

Dear Dr. Asante-Afari,

Thank you for submitting your manuscript to PLOS Global Public Health. After careful consideration, we feel that it has merit but does not fully meet PLOS Global Public Health’s publication criteria as it currently stands. Therefore, we invite you to submit a revised version of the manuscript that addresses the points raised by the reviewers. 

We look forward to receiving your revised manuscript.

Kind regards,

Mohammad Bellal Hossain, PhD

Academic Editor

Journal Requirements:

1. Please change the article type to 'Research Article', and ensure all submission questions are completed, noting the change in publication fees associated with Research Articles. 

Additional Editor Comments (if provided):

Reviewers' comments:

Reviewer's Responses to Questions

**Comments to the Author**

Reviewer #1: All comments have been addressed

Reviewer #3: (No Response)

Reviewer #4: All comments have been addressed

Reviewer #5: (No Response)

publication criteria?

Reviewer #1: Yes

Reviewer #3: Yes

Reviewer #4: Partly

Reviewer #5: Yes

3. Has the statistical analysis been performed appropriately and rigorously?

Reviewer #1: Yes

Reviewer #3: Yes

Reviewer #4: Yes

Reviewer #5: N/A

4. Have the authors made all data underlying the findings in their manuscript fully available (please refer to the Data Availability Statement at the start of the manuscript PDF file)?

Reviewer #1: Yes

Reviewer #3: Yes

Reviewer #4: Yes

Reviewer #5: Yes

5. Is the manuscript presented in an intelligible fashion and written in standard English?

Reviewer #1: Yes

Reviewer #3: Yes

Reviewer #4: Yes

Reviewer #5: Yes

Reviewer #1: Dear authors,

I read with great interest the manuscript, which falls within the aim of this Journal. In my honest opinion, the topic is interesting enough to attract the readers’ attention. Nevertheless, authors should clarify some points and improve the discussion, as suggested below. Authors should consider the following recommendations:

In my opinion you have to improve the paper refering in the text to the updated literature on this topic focusing how its really the artifical intelligence in these pts especially in the infertile pathway as well how its important a deep thyroid disfunction in the infertile pathway as well that have usually they nedd to preserv their fertility by freezing their oocyte by vitrification and to perform a NIPT test at beginning of pregnancy as to follow th neonatal Outcomes and Long-Term Follow-Up of Children Born from Frozen Embryo.Its important to focus on the increased risk of post partum hemorraghia treatments.

I suggest you to read and cite these articles:

The Role of Cell and Gene Therapies in the Treatment of Infertility in Patients with Thyroid Autoimmunity

The role of endoglin and its soluble form in pathogenesis of preeclampsia

Open vs. closed vitrification system: which one is safer?

The role of serum potassium and sodium levels in the development of postpartum hemorrhage. A retrospective study

Impact of assisted reproduction techniques on the neuro-psycho-motor outcome of newborns: a critical appraisalNeonatal Outcomes and Long-Term Follow-Up of Children Born from Frozen Embryo, a Narrative Review of Latest Research FindingsCell-Free Fetal DNA and Non-Invasive Prenatal Diagnosis of Chromosomopathies and Pediatric Monogenic Diseases: A Critical Appraisal and Medicolegal Remarks

Assisted Reproductive Techniques and Risk of Congenital Heart Diseases in Children: a Systematic Review and Meta-analysis

Fresh vs. frozen embryo transfer in assisted reproductive techniques: a single center retrospective cohort study and ethical-legal implications

Stem Cells and Infertility: A Review of Clinical Applications and Legal Frameworks

Reviewer #3: An interesting article that is worth publishing after considering my comments.

I reviewed the article and noted several areas for improvement for clarity, coherence, and academic rigor. Here are some key suggestions for review:

General improvements:

Abstract:

Refine the abstract for conciseness and accuracy. Include specific findings rather than general summaries to make it more impactful.

Clearly state the implications of the study for policy or practice.

Introduction:

Clearly articulate the research gap and the specific objectives of the study.

Strengthen the connection between the identified gap and the contribution of the study.

Methodology:

Expand the sampling strategy to clarify how participants were selected and why these groups were selected.

Provide more details about the analysis process, including examples of thematic categories.

Results:

Use tables or graphs to summarize key findings for better visualization.

Include more direct quotes from participants to enrich the narrative and illustrate key themes.

Discussion:

Further discussion by comparing findings with existing literature to highlight similarities and differences.

Address implications of findings in terms of public health policy or future research.

Language and style:

Avoid repetitive phrases, such as “key religious figures,” by using synonyms or rephrasing.

Simplify sentences where possible to improve readability.

Specific suggestions:

Consistency in terminology: Ensure consistent use of terms such as “ART,” “traditional,” and “religious leaders.” Define these terms early in the text if necessary.

Ethical considerations: Expand on how ethical concerns related to ART were addressed, especially regarding participant confidentiality.

Conclusion: Provide a concise summary of the study findings and emphasize their importance. Emphasize practical recommendations for healthcare providers and policymakers.

References:

Ensure that all references are formatted consistently.

Ensure that all citations are accurate and adhere to the journal’s style guide.

Minor edits:

Fix typos and grammatical errors throughout the text (e.g., “Muslims” should be “Muslims”).

Use gender-neutral language where appropriate, such as “participants” instead of “males and females.”

Adjust headings to be more descriptive and reflect the content of each section.

Requests inclusion of several articles in the References and Discussion section:

Forbidden medically-assisted sex selection in Sunni Muslims: A qualitative study. Reproductive Biomedicine Online.

Forbidden medically-assisted sex selection in Sunni Muslim patients: A qualitative study. Reproductive BioMedicine Online,

"What Do Sunni Muslims Think About Religiously Forbidden Reproductive Options?". Human Fertility

Underground gamete donation in Sunni Muslim patients. Journal of religion and health

What do Sunni Muslims think about religiously forbidden reproductive options? Human fertility

Violating Religious Prohibitions to Preserve Family Harmony and Lineage among Sunni Muslims. Marriage & Family Review A

The Ethical Standards of Sunni Muslim Physicians Regarding Fertility Technologies that are Religiously Forbidden. Journal of religion and health

Reviewer #4: Thank you for carrying out such an interesting study. However, I have a few concerns:

1. The author should make sure to cite key statements of facts and not just make them without referencing them.

Example: “With a population of 2.27 million, Accra is a thriving urban centre. The city boasts a significant Christian population, with 83.5% of residents identifying as such, while 11.8% are Muslims.”

2. Can the author clarify whether the composition of the focus group discussions came from the 30 sampled participants or if they recruited participants from outside the 30? If so, the author should kindly make it clear in the manuscript.

3. Can the author specify the number of participants interviewed in English and the number interviewed in Twi?

4. Who translated the Twi interview to English?

5. The author should adopt a single spelling for "Muslims or Moslems."

6. In addition to the author providing the details of the quotes made by the participants, it will be interesting for the author to indicate their age. This will aid readers in understanding the outcomes.

Example (A Muslim father, FGD participant, 34 years old)

7. The authors should include the study's implications and future directions.

8. The authors should pay attention to the referencing style of PLOS Global Public Health.

Reviewer #5: See Attachment

**Do you want your identity to be public for this peer review?** For information about this choice, including consent withdrawal, please see our Privacy Policy

Reviewer #1: **Yes: ** giuseppe gullo

Reviewer #3: No

Reviewer #4: No

Reviewer #5: **Yes: ** Md Shaikh Farid

---

## [Author Response · Author response to Decision Letter 2]

3 Mar 2025

Dear Editor,

Thank you for sending the reviewers’ comments on the manuscript "Exploring the Intersection of Religion and the Acceptance of Assisted Reproductive Technology Practices: An Examination of Selected Religious Groups in Ghana" to me for the opportunity to address them for your consideration. I have addressed all the comments and provided some responses and clarifications where necessary, as per the attached. I have also attached a track-changed and untracked version for your review. I believe the manuscript has improved vastly and look forward to your kind consideration for acceptance and publication.

Thank you.

Kwadwo Asante-Afari

(Correspondent)

---

## [Decision Letter · Decision Letter 2]

30 Apr 2025

PGPH-D-24-01257R2

Religious and Cultural Perspectives on Assisted Reproductive Technology in Ghana: A Comparative Analysis of Traditionalist, Islamic, and Christian Beliefs

Dear Dr. Asante-Afari,

Thank you for submitting your manuscript to PLOS Global Public Health. After careful consideration, we feel that it has merit but does not fully meet PLOS Global Public Health’s publication criteria as it currently stands. Therefore, we invite you to submit a revised version of the manuscript that addresses the points raised during the review process.

The manuscript has been evaluated by two reviewers, and their comments are available below. Please ignore the comments of reviewer 1, which do not seem to refer to your study.

Please do consider the comments made by reviewer 4 and also refer to this reviewer's comments on your previous revision, which were not mentioned in your last "response to reviewers".

Could you please carefully revise the manuscript to address all comments raised?

We look forward to receiving your revised manuscript.

Kind regards,

Steve Zimmerman, PhD

PLOS Staff Editor

Journal Requirements:

Additional Editor Comments (if provided):

Reviewers' comments:

Reviewer's Responses to Questions

**Comments to the Author**

Reviewer #1: All comments have been addressed

Reviewer #4: All comments have been addressed

publication criteria?

Reviewer #1: Partly

Reviewer #4: Yes

3. Has the statistical analysis been performed appropriately and rigorously?

Reviewer #1: Yes

Reviewer #4: Yes

4. Have the authors made all data underlying the findings in their manuscript fully available (please refer to the Data Availability Statement at the start of the manuscript PDF file)?

Reviewer #1: Yes

Reviewer #4: Yes

5. Is the manuscript presented in an intelligible fashion and written in standard English?

Reviewer #1: Yes

Reviewer #4: Yes

Reviewer #1: Dear authors,

I read with great interest the manuscript, which falls within the aim of this Journal. In my honest opinion, the topic is interesting enough to attract the readers’ attention. Nevertheless, authors should clarify some points and improve the discussion, as suggested below. Authors should consider the following recommendations:

Manuscript should be further revised in order to correct some typos and improve style.

Accumulating evidence suggests that several vitamin dysbalances, as well as nutraceutical supplementation to counteract them, may play a significant role on women’s health especially in breast cancer prevention as well in puberty Since the importance of the topic, I would discuss, at least briefly, adding more details.

I suggest you to read and cite these articles:

An association of boswellia, betaine and myo-inositol (Eumastós®) in the treatment of mammographic breast density: A randomized, double-blind study and quality of life in high-performance swimmers: An observational study

The Role of Cell and Gene Therapies in the Treatment of Infertility in Patients with Thyroid Autoimmunity

Does Alpha-lipoic acid improve effects on polycystic ovary syndrome?

Inositols administration: further insights on their biological role

Oncofertility and Reproductive Counseling in Patients with Breast Cancer: A Retrospective Study

Impact of lifestyle and diet on endometriosis: a fresh look to a busy corner

Myo-inositol: from induction of ovulation to menopausal disorder management

Reviewer #4: The work conducted by the authors is intriguing, but I have some concerns:

1. In the introduction section, the author(s) indicate that “In Ghana, approximately 15% of women of childbearing age are estimated to experience infertility (Donkor & Sandall, 2009).”

That was in 2009; however, what is the current data saying about the women of childbearing age estimated to experience infertility in Ghana?

2. Kindly provide citations for the facts presented in the section titled “setting.”

3. The authors must also explain how they ensured a fair representation of the study's participants, even though they used purposive sampling.

4. The author(s) indicate that the interviews were conducted in English and Twi. Can they indicate the number of interviews carried out in English and those carried out in Twi?

5. Also, in what language was the focus group discussion carried out?

6. Were the participants in the focus group selected from the initial sample? If so, could you provide information about their background, including whether they were Islamic, Christian, or traditional? Only stating the numbers is not enough.

7. Who translated the interview in Twi?

8. The author(s) should clearly indicate the implications of their study.

9. The author(s) should add the interview questions as an appendix.

**Do you want your identity to be public for this peer review?** For information about this choice, including consent withdrawal, please see our Privacy Policy

Reviewer #1: **Yes: ** giuseppe gullo

Reviewer #4: No

---

## [Decision Letter · Decision Letter 3]

9 Jun 2025

PGPH-D-24-01257R3

Religious and Cultural Perspectives on Assisted Reproductive Technology in Ghana: A Comparative Analysis of Traditionalist, Islamic, and Christian Beliefs

Dear Dr. Asante-Afari,

Thank you for submitting your manuscript to PLOS Global Public Health. After careful consideration, we feel that it has merit but does not fully meet PLOS Global Public Health’s publication criteria as it currently stands. Therefore, we invite you to submit a revised version of the manuscript that addresses the points raised during the review process.

We look forward to receiving your revised manuscript.

Kind regards,

Nancy Angeline Gnanaselvam

Academic Editor

**Comments from PLOS Editorial Office:** We note that one or more reviewers has recommended that you cite specific previously published works. As always, we recommend that you please review and evaluate the requested works to determine whether they are relevant and should be cited. It is not a requirement to cite these works. We appreciate your attention to this request.

Additional Editor Comments (if provided):

Kindly follow COREQ checklist in your manuscript writing

Reflexivity statement is required

Authors are required to draw upon principles of human dignity in understanding the themes that emerged from the study. Dignitas infinita is a useful document to refer to

Conceptual framework diagram is required for better understanding

Authors are requested to bring in the social construct of gender and patriarchy in understanding the statements mentioned by participants

Reviewers' comments:

Reviewer's Responses to Questions

**Comments to the Author**

Reviewer #1: All comments have been addressed

publication criteria?

Reviewer #1: Partly

3. Has the statistical analysis been performed appropriately and rigorously?

Reviewer #1: Yes

4. Have the authors made all data underlying the findings in their manuscript fully available (please refer to the Data Availability Statement at the start of the manuscript PDF file)?

Reviewer #1: Yes

5. Is the manuscript presented in an intelligible fashion and written in standard English?

Reviewer #1: Yes

Reviewer #1: I read with great interest the manuscript, which falls within the aim of this Journal. In my honest opinion, the topic is interesting enough to attract the readers’ attention. Nevertheless, authors should clarify some points and improve the discussion, as suggested below. Authors should consider the following recommendations:

In my opinion you have to improve the paper refering in the text to the updated literature on this topic focusing how its really important to preserv their fertility before a fertility sparing treatment for future use of their oocyte by vitrification oocyte as for oncological reasons for gynecological malignancies as endometrail or breast cancer

focusing how its really important to perform a long term follow up of these babies born from frozen embryos anfter IVF.

I suggest you to read and cite these articles:

Oncofertility and Reproductive Counseling in Patients with Breast Cancer: A Retrospective Study

Neonatal Outcomes and Long-Term Follow-Up of Children Born from Frozen Embryo, a Narrative Review of Latest Research Findings

Fertility-Sparing Strategies for Early-Stage Endometrial Cancer: Stepping towards Precision Medicine Based on the Molecular Fingerprint

Impact of assisted reproduction techniques on the neuro-psycho-motor outcome of newborns: a critical appraisal

The era of increasing cancer survivorship: Trends in fertility preservation, medico-legal implications, and ethical challenges

Closed vs. Open Oocyte Vitrification Methods Are Equally Effective for Blastocyst Embryo Transfers: Prospective Study from a Sibling Oocyte Donation Program

Fresh vs. frozen embryo transfer in assisted reproductive techniques: a single center retrospective cohort study and ethical-legal implications

**Do you want your identity to be public for this peer review?** For information about this choice, including consent withdrawal, please see our Privacy Policy

Reviewer #1: **Yes: ** GIUSEPPE gullo

---

## [Decision Letter · Decision Letter 4]

8 Aug 2025

PGPH-D-24-01257R4

Religious and Cultural Perspectives on Assisted Reproductive Technology in Ghana: A Comparative Analysis of Traditionalist, Islamic, and Christian Beliefs

Dear Dr. Asante-Afari,

Thank you for submitting your manuscript to PLOS Global Public Health. After careful consideration, we feel that it has merit but does not fully meet PLOS Global Public Health’s publication criteria as it currently stands. Therefore, we invite you to submit a revised version of the manuscript that addresses the points raised during the review process.

We look forward to receiving your revised manuscript.

Kind regards,

Nancy Angeline Gnanaselvam

Academic Editor

Journal Requirements:

Additional Editor Comments:

Please address the reviewers comments

Reviewers' comments:

Reviewer's Responses to Questions

**Comments to the Author**

Reviewer #6: All comments have been addressed

publication criteria?

Reviewer #6: Yes

3. Has the statistical analysis been performed appropriately and rigorously?

Reviewer #6: Yes

4. Have the authors made all data underlying the findings in their manuscript fully available (please refer to the Data Availability Statement at the start of the manuscript PDF file)?

Reviewer #6: No

5. Is the manuscript presented in an intelligible fashion and written in standard English?

Reviewer #6: Yes

Reviewer #6: Approves the article for publication after addressing the reviewer's comments:

The following is a positive response to the article, along with minor recommendations for improvement regarding statistical clarification, polishing the English language, and expanding the coverage of religious perspectives - especially those noted by Ya'arit Bokek.

Reviewer's Comments - Minor Correction / Final Feedback

Overall Evaluation:

I commend the author for a well-structured and informative manuscript, which explores a highly relevant and sensitive topic - religious and cultural perspectives on assisted reproductive care in Ghana. The article provides valuable insights into the diverse beliefs and practices of traditional, Islamic, and Christian communities and how these influence the perception and acceptance of assisted reproductive technologies.

Having reviewed the revised manuscript and the responses to the previous reviewer's comments, I am pleased to approve it for publication after addressing the minor comments below. These suggestions are intended to strengthen clarity, enhance academic rigor, and improve the presentation of the manuscript.

Statistical and methodological clarification

Clarification of participant distribution: While the demographic breakdown in Table 2 is helpful, the qualitative sample of 30 participants could benefit from a clearer explanation of the rationale behind its size and how thematic saturation was determined. A brief mention of the principles guiding saturation in qualitative research (e.g., Guest et al., 2006) would strengthen methodological transparency.

Duration and timing: The recruitment period (March–August 2017) should be briefly addressed in the discussion—consider adding a sentence explaining whether these positions have changed in recent years and how the passage of time affects current applicability.

Language and style adjustments in English

While the manuscript is generally well written, I recommend a final professional proofreading to improve clarity. Here are some sample corrections and stylistic suggestions:

Summary:

Source: "...while respecting people's religious beliefs and values."

Suggested: "...while respecting people's religious beliefs and values."

Page 13:

Source: "...this act is outright prostitution..."

Suggested: "...this act is akin to prostitution..."

(To maintain an academic tone(

Page 17:

Source: "Why freeze your eggs or sperm?"

Suggested: "What is the purpose of freezing a person's eggs or sperm?"

(For clarity and professionalism.)

General: The use of terms such as "dirty," "illegitimate," and "abomination" appears in direct quotes from participants. Consider marking them as culturally or contextually situated in the analysis to avoid unintentional offense or misinterpretation.

Commentary on Religious Frameworks - Including Bokek's Contribution

The manuscript addresses religious perspectives well, but the excellent contributions of Ya'arit Bokek-Cohen and Tarabiyeh (2021, 2022) are only briefly mentioned. Given the direct relevance of their findings to the Islamic and Christian ethics of antiretroviral therapy, I highly recommend the following:

Violating Religious Prohibitions to Preserve Family Harmony and Lineage among Sunni Muslims. Marriage & Family Review

What do Sunni Muslims think about religiously forbidden reproductive options?. Human fertility

Underground gamete donation in Sunni Muslim patients. Journal of religion and health

The Ethical Standards of Sunni Muslim Physicians Regarding Fertility Technologies that are Religiously Forbidden

Additional Integration: The studies of Bokek-Cohen et al. (2021, 2022) should be more thoroughly integrated into the discussion of Sunni Muslim views, especially regarding:

The secretive nature of gamete donation.

The tension between personal fertility desires and communal/religious prohibitions.

Gender dynamics in the deception and secrecy of partners in the use of antiretroviral therapy.

Comparative Insight: A few lines contrasting the Ghanaian religious context with other Muslim or Christian societies (e.g., Israel, Egypt, or Turkey) could enrich the discussion and global applicability.

This literature by Bokek-Cohen et al. may be referenced, as well as other literature if any:

Minor recommendations for future research

Future work should consider including:

Perspectives from young unmarried adults, who may represent changing generational perspectives.

The legal and policy implications of antiretroviral treatment practices in Ghana, particularly around inheritance, adoption, and custody laws.

Add a brief note in the conclusion on how public health initiatives might engage religious leaders in designing more inclusive antiretroviral drug awareness campaigns.

Conclusion:

This article is an important contribution to the literature on religion and antiretroviral drug treatment in West Africa. With some refinement and expanded attention to comparative literature—particularly regarding Islamic jurisprudence and Christian theological debates—the manuscript would be a valuable resource for both academic and policy audiences.

I support its publication after minor correction.

**Do you want your identity to be public for this peer review?** For information about this choice, including consent withdrawal, please see our Privacy Policy

Reviewer #6: **Yes: ** Approves the article for publication after addressing the reviewer's comments:

The following is a positive response to the article, along with minor recommendations for improvement regarding statistical clarification, polishing the English language, and expanding the coverage of religious perspectives - especially those noted by Ya'arit Bokek.

Reviewer's Comments - Minor Correction / Final Feedback

Overall Evaluation:

I commend the author for a well-structured and informative manuscript, which explores a highly relevant and sensitive topic - religious and cultural perspectives on assisted reproductive care in Ghana. The article provides valuable insights into the diverse beliefs and practices of traditional, Islamic, and Christian communities and how these influence the perception and acceptance of assisted reproductive technologies.

Having reviewed the revised manuscript and the responses to the previous reviewer's comments, I am pleased to approve it for publication after addressing the minor comments below. These suggestions are intended to strengthen clarity, enhance academic rigor, and improve the presentation of the manuscript.

Statistical and methodological clarification

Clarification of participant distribution: While the demographic breakdown in Table 2 is helpful, the qualitative sample of 30 participants could benefit from a clearer explanation of the rationale behind its size and how thematic saturation was determined. A brief mention of the principles guiding saturation in qualitative research (e.g., Guest et al., 2006) would strengthen methodological transparency.

Duration and timing: The recruitment period (March–August 2017) should be briefly addressed in the discussion—consider adding a sentence explaining whether these positions have changed in recent years and how the passage of time affects current applicability.

Language and style adjustments in English

While the manuscript is generally well written, I recommend a final professional proofreading to improve clarity. Here are some sample corrections and stylistic suggestions:

Summary:

Source: "...while respecting people's religious beliefs and values."

Suggested: "...while respecting people's religious beliefs and values."

Page 13:

Source: "...this act is outright prostitution..."

Suggested: "...this act is akin to prostitution..."

(To maintain an academic tone(

Page 17:

Source: "Why freeze your eggs or sperm?"

Suggested: "What is the purpose of freezing a person's eggs or sperm?"

(For clarity and professionalism.)

General: The use of terms such as "dirty," "illegitimate," and "abomination" appears in direct quotes from participants. Consider marking them as culturally or contextually situated in the analysis to avoid unintentional offense or misinterpretation.

Commentary on Religious Frameworks - Including Bokek's Contribution

The manuscript addresses religious perspectives well, but the excellent contributions of Ya'arit Bokek-Cohen and Tarabiyeh (2021, 2022) are only briefly mentioned. Given the direct relevance of their findings to the Islamic and Christian ethics of antiretroviral therapy, I highly recommend the following:

Violating Religious Prohibitions to Preserve Family Harmony and Lineage among Sunni Muslims. Marriage & Family Review

What do Sunni Muslims think about religiously forbidden reproductive options?. Human fertility

Underground gamete donation in Sunni Muslim patients. Journal of religion and health

The Ethical Standards of Sunni Muslim Physicians Regarding Fertility Technologies that are Religiously Forbidden

Additional Integration: The studies of Bokek-Cohen et al. (2021, 2022) should be more thoroughly integrated into the discussion of Sunni Muslim views, especially regarding:

The secretive nature of gamete donation.

The tension between personal fertility desires and communal/religious prohibitions.

Gender dynamics in the deception and secrecy of partners in the use of antiretroviral therapy.

Comparative Insight: A few lines contrasting the Ghanaian religious context with other Muslim or Christian societies (e.g., Israel, Egypt, or Turkey) could enrich the discussion and global applicability.

This literature by Bokek-Cohen et al. may be referenced, as well as other literature if any:

Minor recommendations for future research

Future work should consider including:

Perspectives from young unmarried adults, who may represent changing generational perspectives.

The legal and policy implications of antiretroviral treatment practices in Ghana, particularly around inheritance, adoption, and custody laws.

Add a brief note in the conclusion on how public health initiatives might engage religious leaders in designing more inclusive antiretroviral drug awareness campaigns.

Conclusion:

This article is an important contribution to the literature on religion and antiretroviral drug treatment in West Africa. With some refinement and expanded attention to comparative literature—particularly regarding Islamic jurisprudence and Christian theological debates—the manuscript would be a valuable resource for both academic and policy audiences.

I support its publication after minor correction.

---

## [Editor Report · Decision Letter 5]

7 Sep 2025

Religious and Cultural Perspectives on Assisted Reproductive Technology in Ghana: A Comparative Analysis of Traditionalist, Islamic, and Christian Beliefs

PGPH-D-24-01257R5

Dear Dr Asante-Afari,

We are pleased to inform you that your manuscript 'Religious and Cultural Perspectives on Assisted Reproductive Technology in Ghana: A Comparative Analysis of Traditionalist, Islamic, and Christian Beliefs' has been provisionally accepted for publication in PLOS Global Public Health.

Best regards,

Nancy Angeline Gnanaselvam

Academic Editor

Author has responded to reviewers' queries satisfactorily. The paper reads well. It gives a diverse perspective on family and health. It will encourage more global health research on intersection of humanities and health